# Ecological Stoichiometric Characteristics of Plant–Soil–Microorganism of Grassland Ecosystems under Different Restoration Modes in the Karst Desertification Area

Shuzhen Song, Kangning Xiong * and Yongkuan Chi

School of Karst Science, Engineering Laboratory for Karst Desertification Control and Eco-Industry of Guizhou Province, Guizhou Normal University, Guiyang 550001, China; 15010170764@gznu.edu.cn (S.S.); 201907002@gznu.edu.cn (Y.C.)
* Correspondence: xiongkn@gznu.edu.cn

**Abstract:** C, N and P are the key biogenic elements of terrestrial ecosystems, and their biogeochemical processes regulate nutrient cycling and play a key role in restoring degraded ecosystems. In this paper, the grassland ecosystem under artificial restoration measures (*Dactylis glomerata* (DG), *Lolium perenne* (LP), *Lolium perenne + Trifolium repens* (LT)), and the natural restoration measures (NG) in a typical karst plateau desertification control area of southwest China, were taken as the research object. The C, N, and P concentrations and the ecological stoichiometry of the plant–soil–microorganism system in grasslands under different restoration measures in the karst desertification area were explored. We established the following findings: (1) Compared with NG, the C, N and P concentrations of plants and soil in DG, LP and LT were higher, and LT was the highest. (2) The microbial biomass carbon (MBC), microbial biomass nitrogen (MBN) and microbial biomass phosphorus (MBP) concentrations in LT was also the highest. (3) The C:N ratio of plant and soil indicated that the N mineralization and nutrient release rate of DG, LP and LT were stronger than that of NG, and the plant growth of NG was most seriously limited by nitrogen. The N:P ratio in plant and soil indicated that the grassland was limited by P in the four treatments. (4) The result of correlation analysis showed that the cycling process of C, N and P in the plant–soil–microorganism system was coupled, and that the elements closely affected each other. In general, the effect of artificial restoration on a degraded ecosystem was relatively better than the natural restoration for increasing plant, soil and microbial nutrient concentrations, especially in the mixed-seed ecosystem of leguminous and gramineous forage. However, in the future, grassland management, appropriate N fertilizer or N-fixing plants and P fertilizer should be considered to improve the limitation of plant N and P, so as to realize the sustainable development of grasslands and the restoration of degraded ecosystems in the karst desertification control area.

**Keywords:** plant–soil–microorganism system; ecological stoichiometry; artificial grassland; natural restoration; karst desertification

## 1. Introduction

Carbon, nitrogen and phosphorus are the key biogenic elements of the terrestrial ecosystem, among which N and P are two kinds of essential macronutrient elements in the plant life cycle, and their content can affect plant community composition and productivity levels; meanwhile, their biogeochemical processes play a key role in nutrient cycles, ecosystem processes and functioning [1,2]. Plants, soils and microorganisms are closely related and interact with each other. Soil microorganisms release nutrient elements by decomposing soil organic matter and litter, form and improve soil structure and nutrient composition and finally regulate plant growth, while plant litter can return part of the nutrients to the soil for soil microorganism decomposition [3]. Due to the complexity of and differences between key ecosystem processes such as plant community composition,

soil nutrient turnover, microbial community structure, etc., significant differences in plant–soil–microbial C, N and P contents are inevitable [4,5]. The cycle and accumulation process of nutrients among plants, soils and microorganisms promotes the restoration, productivity and stability of ecosystems. So, it is helpful to reveal nutrient cycling processes and mechanisms of stability by studying the internal relationship of ecosystem processes and their feedback effects with C, N and P as carriers in plant–soil–microorganism systems [6,7].

Ecological stoichiometry, as an important method to study the balance of the main elements and material cycles of each component of the ecosystem, provides an important framework for research related to biogeochemical cycles [6], especially in plant growth, element limitation and ecosystem stability [7]. As an important component of terrestrial ecosystems, soil is the carrier connecting many ecological processes of ecosystems, affecting the composition and stability of plant community and dynamic changes in soil microorganisms [8]. The nutrient elements of plants, soil and microorganisms are coupled in the ecosystem process, but it is difficult to reveal the ecosystem process and quality change only through the study of nutrient elements, while C:N:P ratios can reflect the functional characteristics of nutrient use efficiency and nutrient limitation [9–11]. Currently, there are numerous studies on the ecological stoichiometry characteristics of degraded ecosystems, involving a wide range of research areas and objects [12–16], and these studies showed that there were certain differences in C, N, and P concentrations and ecological stoichiometry of plant–soil–microorganism systems in different regions, habitats and vegetation restoration measures. Vegetation restoration measures are one of the important factors affecting the C, N and P cycle of plant–soil–microorganism systems [17]. Different vegetation restoration measures will break the balance of elements and substances in the original structure, thus leading to changes in the concentration and ecological stoichiometry of C, N and P [18,19]. Therefore, it is necessary to study the concentrations of C, N and P in plant–soil–microorganism systems and the ecological stoichiometry under different restoration measures in order to reveal the coupling relationship between nutrients and changes in ecosystem processes, especially for the restoration of degraded ecosystems [9,11,12].

The South China Karst is centered on the Guizhou plateau, which is the largest and most contiguous fragile karst area among the three extensive karst distribution areas in the world, with a total area of more than $55 \times 10^4$ km$^2$ [20]. Influenced by its geological background and long-term human factors, rocky desertification is the most serious eco-environmental problem in this region [21]. Rocky desertification is a landscape phenomenon in which karst landscapes in tropical and subtropical climates are disturbed by the natural environment and humans, resulting in severe soil erosion, vegetation decline and a desert-like surface [21]. Among them, Guizhou is the most typical karst province in China and even in the world, its karst areas and rocky desertification areas accounting for 73% and 21.34% of the total land area of Guizhou, respectively, and rocky desertification control is the main task and difficulty of ecological restoration [22]. Studies have shown that "grain for green" projects in rocky desertification areas are an important measure to restore the ecological environment damaged by rocky desertification, which is of great significance to promote ecological restoration and economic development [23]. The establishment of artificial grassland or natural recovery and secondary succession without human intervention of grassland may change the soil–plant–microorganism nutrient relationship formed by long-term evolution [23,24], thus having a profound impact on the balance of elements and substances in the ecosystem [25,26]. Up to now, research on ecological stoichiometry in karst areas has seen many achievements in vegetation succession, some vegetation (e.g., forest) or areas (e.g., grassland in northern China) [27–31], but research on ecological stoichiometry of the plant–soil–microorganism system in grasslands of the karst desertification control area is still very weak. At the same time, research on the C, N and P concentration of plant–soil–microorganism and ecological stoichiometry characteristics of grassland ecosystems under different restoration measures in the karst desertification control area is still insufficient.

Therefore, this study was carried out in the grassland ecosystem under artificial restoration measures (*Dactylis glomerata* (DG), *Lolium perenne* (LP), *Lolium perenne + Trifolium repens* (LT)) and natural restoration measures (NG) in a typical karst plateau rocky desertification control area of southwest China as the research object. The objectives were: (1) to investigate the characteristics of C, N and P concentrations in plants, soils and microorganisms of grassland ecosystems under different restoration methods; (2) to analyze the ecological stoichiometry of C, N and P in plants, soils and microorganisms of the grassland ecosystem under different restoration methods; and (3) to explore the relationship between the C, N and P concentrations and ecological stoichiometry of plants, soils and microorganisms of grassland ecosystems under different restoration methods, so as to provide a scientific basis for the sustainable utilization of grasslands and ecological restoration in the karst desertification control area.

## 2. Materials and Methods

### 2.1. Study Area

The study area is located in the upper reaches of the Yangtze River Liuchonghe River basin in Yunnan-Guizhou Plateau, Qixingguan District, Bijie City, Guizhou Province, China (105°02′01″–105°08′09″ E, 27°11′36″–27°16′51″ N), which has an altitude of about 1800 m (Figure 1). It is a karst plateau mountain area with potential light desertification; the karst desertification area is 55.931 km$^2$, accounting for 64.9% of the study area. The study area is a subtropical monsoon humid climate. The average annual temperature is about 12 °C, the frost-free period is 245 days, the average annual sunshine hours are 1360 h and the average annual precipitation is 984.4 mm. The soil type is mainly yellow soil. The existing tree-shrub vegetation in this study area mainly consists of *Pinus yunnanensis*, *Betula luminifera*, *Cyclobalanopsis glauca*, *Rhododendron simsii*, *Pyracantha fortuneana*, *Rubus corchorifolius*, *Rosa roxburghii*, *Rhus chinensis* and *Juglans regia*. Herbaceous plants include *Plantago depressa*, *Artemisia caruifolia*, *Chenopodium glaucum*, *Stellaria media*, *Digitaria sanguinalis*, *Trifolium repens*, *Lolium perenne*, *Dactylis glomerata*, *Bromus catharticus* and *Festuca elata*.

### 2.2. Sample Plot Setting

According to the layout of the rocky desertification control project, and considering the development of herbivorous animal husbandry, the research group established artificial grassland and returned farmland to grassland in the study area in 2012 [21]. In this study, all land was planted with corn before planting grass and natural restoration. After the implementation of the rocky desertification control project, the land uses mode was changed from corn planting to artificial grass establishment or returning farmland to grassland (natural restoration). The established forage included *Trifolium repens*, *Lolium perenne*, *Dactylis glomerata*, *Bromus catharticus* and *Festuca elata*. The establishment of artificial grassland included single seeding and mixed seeding. Therefore, the grassland ecosystem in this study area included a natural grassland ecosystem and an artificial grassland ecosystem; meanwhile, the artificial grassland ecosystem included a single-seed grassland ecosystem and a mixed-seed grassland ecosystem. In the study area, three grassland types were selected as the experimental plots, i.e., *Dactylis glomerata* (DG), *Lolium perenne* (LP) and *Lolium perenne + Trifolium repens* (LT), and the natural grassland (NG) was used as the control (Figure 1). The variety of *Trifolium repens* was "Haifa", the variety of *Dactylis glomerata* was "Qiangrass No. 4" and the variety of *Lolium perenne* was "Yaqing", and the three kinds of forage seeds were provided by Guizhou Shennong Seed Industry Co., Ltd. (Guiyang, China) and Lvyi Seed Industry Co., Ltd. (Guiyang, China). No agricultural activities or human disturbances were carried out in the natural restoration grassland. The annual dry weight of DG, LP, LT and NG was 1049 g/m$^2$, 1593 g/m$^2$, 1988 g/m$^2$ and 1153 g/m$^2$, respectively.

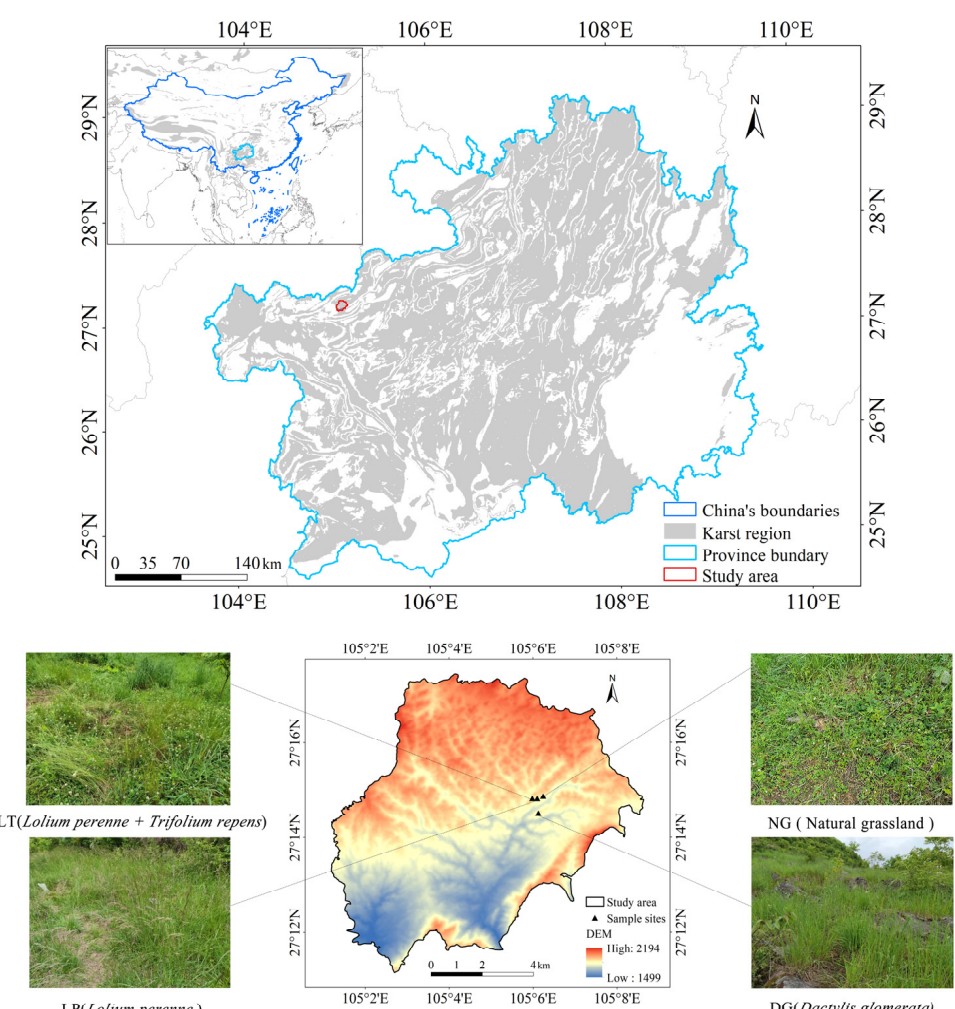

**Figure 1.** Study area and sampling sites location.

### 2.3. Sample Collection

In the middle of July 2021, 6 sampling plots of 10 m × 10 m (24 sampling plots in total) were set in each of the four restoration treatments, and the distance and the boundary of sample plot was greater than 10 m. Five 1 m × 1 m quadrats were set in each sampling plot for sample collection. All plants in each 1 m × 1 m quadrat were clipped and mixed into one replicate sample; a total of 24 plant samples were obtained. Soil samples were collected from the surface layer (0–20 cm) using a soil auger at the same point, and soil samples from 5 quadrats were mixed into one replicate sample, and 24 soil samples were obtained. After removing the rocks and large organic debris in the soil sample, the sample was divided into two parts. One was naturally air-dried in the laboratory, and then passed through a 2 mm sieve for the determination of soil carbon (SOC), nitrogen (TN) and phosphorus (TP) concentrations. The other was stored in a refrigerator at −80 °C for the determination of soil microbial biomass carbon, nitrogen and phosphorus. The plant samples were brought back to the laboratory and next inactivated at 105 °C for 30 min and then dried at 75 °C for 48 h to a constant weight. After drying, plant samples were normally ground to a fine powder using a ball grinder and sieved with a 2 mm sieve for the determination of plant carbon, nitrogen and phosphorus.

### 2.4. Samples Determination

The C and N concentration of plants and soils were determined by an automatic elemental analyzer (FlashSmart, Thermo Fisher, Waltham, WA, USA). The total phosphorus (TP) concentration of plants was digested with $H_2SO_4$-$H_2O_2$, and the soil was fused

with NaOH and colored with molybdenum-antimony resistance, and then determined with ultraviolet spectrophotometer (Specord 200 PLUS, Analytik, Jena, Germany). Soil microbial biomass carbon (MBC), microbial biomass nitrogen (MBN) and microbial biomass phosphorus (MBP) were determined using the $CHCl_3$ fumigation extraction method [32].

*2.5. Data Analysis*

One-way analysis of variance, multiple comparison with least significant difference and Pearson correlation analysis were performed in SPSS (version 19.0 for Windows; SPSS, Chicago, IL, USA) to analyze the effects of different restoration modes on ecological stoichiometry of soil–plant–microorganism system in grasslands, and the figures were plotted using Origin (version 2021 for Windows; OriginLab, Northampton, MA, USA).

## 3. Results

*3.1. C, N and P Concentrations and Ecological Stoichiometry of Plants in Grassland Ecosystems*

The C, N and P concentrations, and their ecological stoichiometry, of plants in grassland ecosystems under the four restoration modes were quite different (Figure 2). The C concentration of plant was LT > DG > LP > NG, and the difference between NG and DG, and LP and LT, was significant, but the difference among DG, LP and LT was not significant. The N concentration of plants was LT > LP > DG > NG, and LT was significantly higher than the other three treatments, but the difference between NG and DG, and DG and LP was not significant. The P concentration in plants was LT > LP > NG > DG, and LT was significantly higher than the other three treatments, but the difference among DG, LP and LT was not significant.

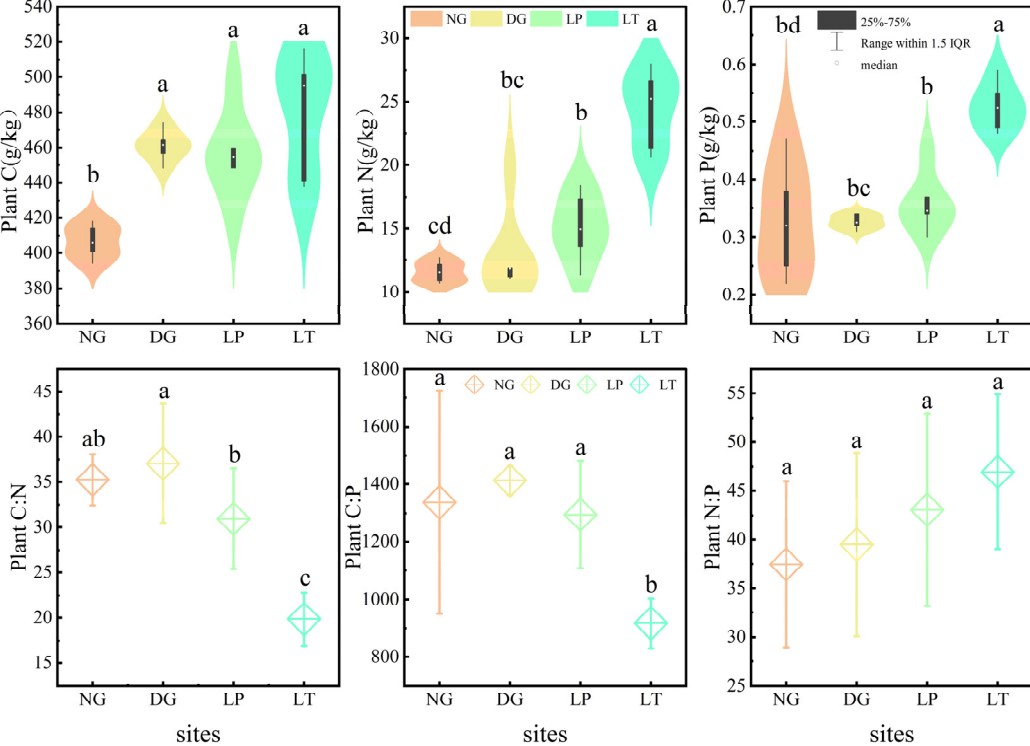

**Figure 2.** C, N and P concentrations and ecological stoichiometry of plants in grassland ecosystem under the different vegetation restoration modes. NG, natural grassland; DG, *Dactylis glomerata*; LP, *Lolium perenne*; LT, *Lolium perenne + Trifolium repens.* Different lowercase letters indicate significance at 0.05 probability level.

The C:N ratio of plants was DG > NG > LP > LT, and the difference between NG and LT was significant, but the difference between DG and LP was not significant. The C:P ratio of plant was DG > NG > LP > LT, and LT was significantly lower than the other

three treatments, but the difference among NG, DG and LP was not significant. The N:P ratio of plant was LT > LP > DG > NG, but there was no significant difference among the four treatments. The mean values of plant C:N, C:P and N:P in the four restoration modes were 30.80, 1240.49 and 41.75, respectively.

### 3.2. C, N and P Concentrations and Ecological Stoichiometry of Soil in Grassland Ecosystems

The C, N and P concentrations and their ecological stoichiometry of soil in grassland ecosystems under the four restoration modes were quite different (Figure 3). The SOC and TN concentrations of soil were LT > LP > DG > NG, and LT was significantly higher than the other three treatments, but the difference between DG and LP was not significant. The TP concentration of soil was LT > LP > NG > DG, and LT was significantly higher than the other three treatments, but the difference between NG and DG was not significant. The mean values of SOC, TN and TP of soil in the four restoration modes were 18.49 g kg$^{-1}$, 1.53 g kg$^{-1}$ and 1.06 g kg$^{-1}$, respectively.

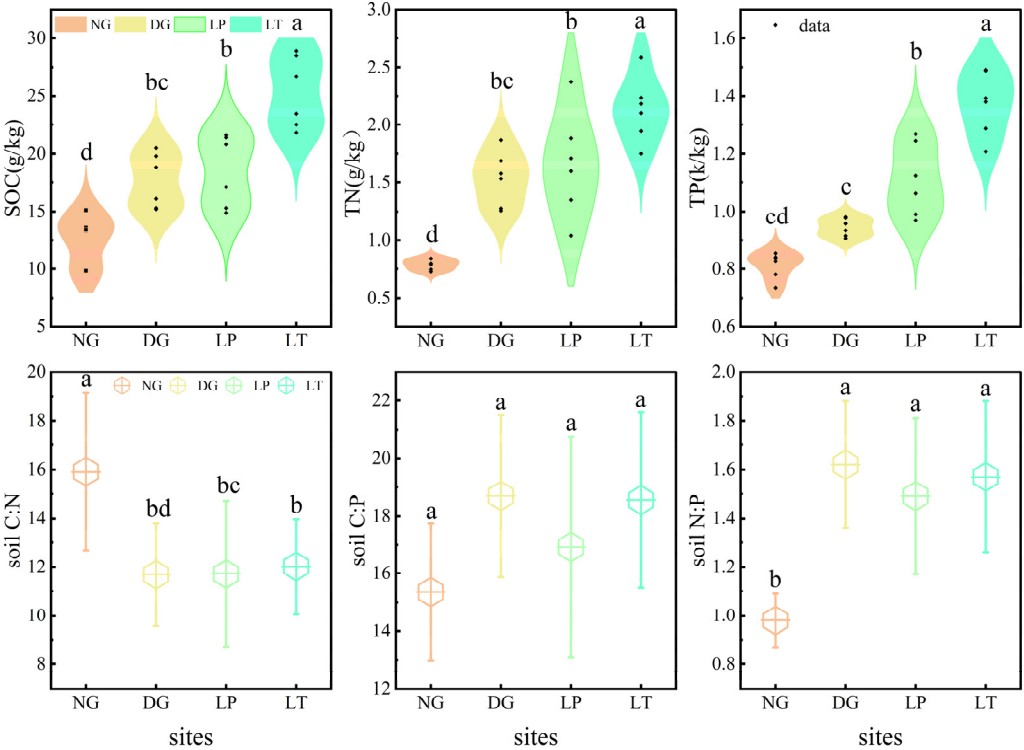

**Figure 3.** C, N and P concentrations and ecological stoichiometry of soil in grassland ecosystems under the different vegetation restoration modes. NG, natural grassland; DG, *Dactylis glomerata*; LP, *Lolium perenne*; LT, *Lolium perenne + Trifolium repens*. Different lowercase letters indicate significance at 0.05 probability level.

The C:N ratio of soil was NG > LT > LP > DG, and the difference between NG and other three treatments was significant, but the difference among DG, LP and LT was not significant. The C:P ratio of soil was DG > LT > LP > NG, and the difference among the four treatments was not significant. The N:P ratio of soil was LT > DG > LP > NG, and NG was significantly lower than the other three treatments, but the difference among DG, LP and LT treatments was not significant. The mean values of soil C:N, C:P and N:P in the four restoration modes were 12.83, 17.38 and 1.42, respectively.

### 3.3. MBC, MBN and MBP Concentrations and Ecological Stoichiometry of Grassland Ecosystems

The concentrations of MBC, MBN and MBP and their ecological stoichiometry of grassland ecosystems under the four restoration modes were quite different (Figure 4). The MBC concentration was LT > LP > DG > NG, and LT was significantly higher than

the other three treatments, but the difference between NG, LP and DG was not significant. The MBN concentration was LT > LP > DG > NG, and LT was higher than the other three treatments, but the difference between DG and LP was not significant. The MBP concentration was LT > LP > DG > NG, and the differences among the four treatments were significant; the LT was the highest and the NG was the lowest. The average values of MBC, MBN and MBP in the four restoration modes were 318.37 mg kg$^{-1}$, 92.64 mg kg$^{-1}$ and 15.61 mg kg$^{-1}$, respectively.

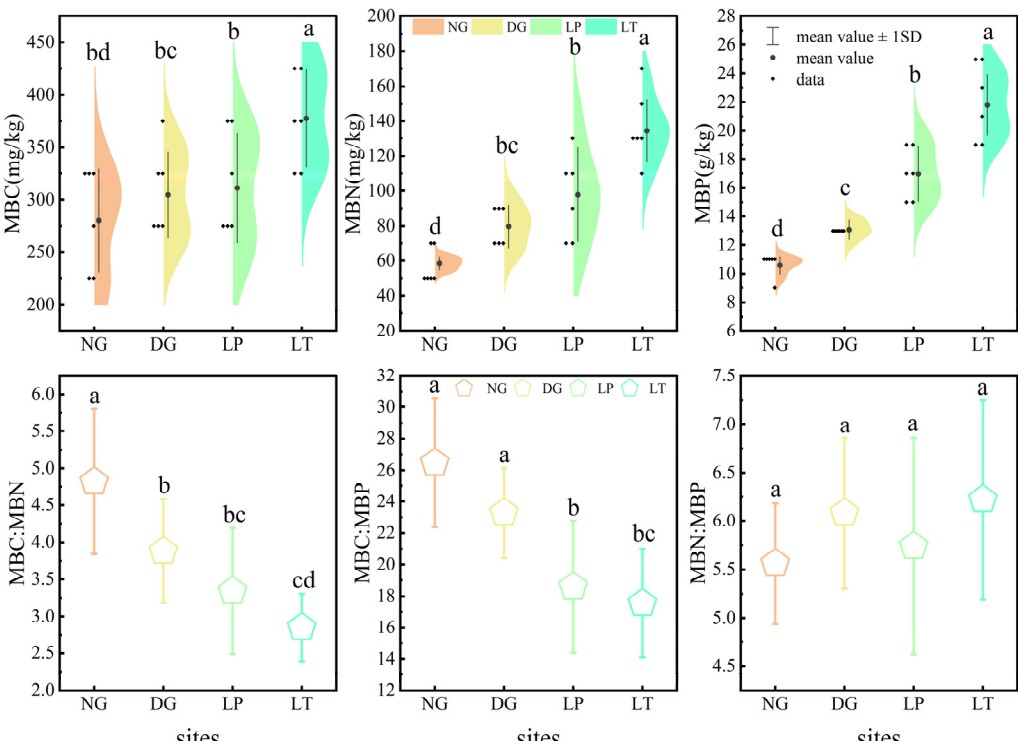

**Figure 4.** MBC, MBN and MBP concentrations and stoichiometric ratios in grassland ecosystems under different vegetation restoration modes. NG, natural grassland; DG, *Dactylis glomerata*; LP, *Lolium perenne*; LT, *Lolium perenne + Trifolium repens.* Different lowercase letters indicate significance at 0.05 probability level.

MBC:MBN showed NG > DG > LP > LT, and NG was significantly higher than the other three treatments, but the difference between DG and LP, and LP and LT, was not significant. MBC:MBP showed NG > DG > LP > LT, and NG was significantly higher than LP and LT, but not significantly higher than DG. MBN:MBP showed LT > DG > LP > NG, and the difference among the four treatments was not significant. The mean values of MBC:MBN, MBC:MBP and MBN:MBP in the four restoration modes were 3.72, 21.47 and 5.90, respectively.

*3.4. Correlation Analysis of C, N and P Concentrations and Ecological Stoichiometry of Plant–Soil–Microbial Interactions in Grassland Ecosystems*

The correlation analysis results (Figure 5) showed that the positive correlation of SOC was highly significant with TN, TP, plant C, plant N, plant P, MBC, MBN, MBP, soil C:P and soil N:P. There was a highly significant negative correlation with plant C:N, and a significantly negative correlation with plant C:P, MBC:MBN. The positive correlation of TN was highly significant in relation to TP, plant C, plant N, plant P, MBC, MBN, MBP, soil N:P and MBN:MBP. There was a highly significant negative correlation with soil C:N, plant C:N, MBC:MBN and MBC:MBP. The positive correlation of TP was highly significant with plant C, plant N, plant P, MBC, MBN and MBP, and significant with plant N:P. There was

highly negative correlation with plant C:N, MBC:MBN and MBC:MBP, and a significantly negative correlation with plant C:P.

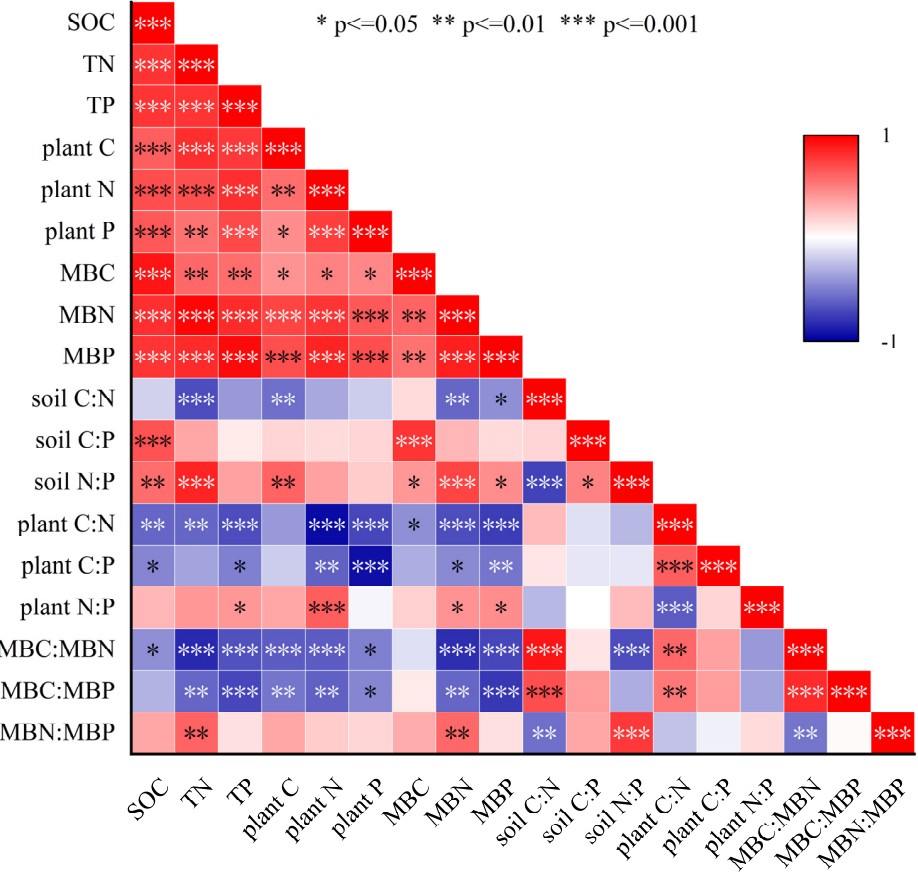

**Figure 5.** Correlation analysis of C, N and P concentrations and ecological stoichiometry in plant–soil–microorganism interactions in grassland ecosystems.

The positive correlation of plant C was highly significant with plant N, MBN, MBP and soil N:P, and significant with plant P, MBC. There was a highly significant negative correlation with soil C:N, MBC:MBN and MBC:MBP. The positive correlation of plant N was highly significant with plant P, MBN, MBP and plant N:P, and significant with MBC. There was a highly significant negative correlation with plant C:N, plant C:P, MBC:MBN and MBC:MBP. The positive correlation of plant P was highly significant with MBN and MBP, and significant with MBC. There was a highly significant negative correlation with plant C:N and plant C:P, and significantly negative correlations with MBC:MBN and MBC:MBP.

The positive correlation of MBC was highly significant with MBN, MBP and soil C:P, and significant with soil N:P. There was a significantly negative correlation with plant C:N. The positive correlation of MBN was highly significant with MBP, soil N:P and MBN:MBP, and significant with plant N:P. There was a highly significant negative correlation with soil C:N, plant C:N, MBC:MBN and MBC:MBP, and significantly negative correlation with plant C:P. The positive correlation of MBP was significant with soil N:P and plant N:P. There was a highly significant negative correlation with plant C:N, plant C:P, MBC:MBN and MBC:MBP, and a significantly negative correlation with soil C:N.

The positive correlation of plant C:N was highly significant with plant C:P, MBC:MBN and MBC:MBP. There was highly significant negative correlation with plant N:P. The positive correlation of soil C:N was highly significant with MBC:MBN and MBC:MBP. There was a highly significant negative correlation with soil N:P and MBN:MBP. Soil C:P was significantly positively correlated with soil N:P. The positive correlation of soil N:P was highly significant with MBN:MBP. There was a highly significant negative correlation with



MBC:MBN. The positive correlation of MBC:MBN was highly significant with MBC:MBP. There was a highly significant negative correlation with MBN:MBP.

## 4. Discussion

### 4.1. Effects of Different Restoration Modes on Plant C, N and P Concentrations and Ecological Stoichiometry in Grassland Ecosystems

C, N and P are basic elements that constitute life, among which N and P are the limiting elements for the growth of plants on land; they play an irreplaceable role in catalyzing metabolic reactions and synthesizing the life components of organisms [33,34]. Plant N and P play an important role in the productivity and C sequestration of terrestrial ecosystems [35]. The biogeochemical cycle of C, N and P in plants are closely related to natural processes such as photosynthesis, respiration and litter decomposition which, in turn, affects the distribution of matter and energy [36]. The plant C concentration of LT in this study was significantly higher than NG, but the difference among DG, LP and LT was not significant, which is consistent with the results of the Inner Mongolia desert steppe area [37]. This may be related to the fact that the primary productivity of artificial grassland is greater than that of natural grassland, which increases the carbon fixation capacity of plants. The functional elements N and P are the main components of the proteins and nucleic acids of living organisms, and they have irreplaceable effects on regulating organisms [38]. The plant N concentration of the artificially restored grassland ecosystem was significantly higher than that with natural restoration, and it was the highest in LT. The main reason for this may be related to the nitrogen fixation characteristics of white clover, which is consistent with the findings of Gao et al. [37]. The plant P concentration was the highest in LT, which indicated that the plant P concentration of mixed–seed gramineous and leguminous grassland was higher than that of single-seed gramineous grassland and naturally restored grassland, which is consistent with the findings of Guo et al. [39].

The C:N and C:P of plants can represent their nutrient utilization efficiency. The larger the value of C:N and C:P, the higher the utilization efficiency of N and P, which is the strategy of plants to deal with soil nutrient deficiency [40]. N:P is an important ecological index to judge the limitation of plant nitrogen and phosphorus nutrition. When N:P is greater than or less than a certain critical value, plant growth is mainly limited by N or P [41]. The C:N and C:P of the four treatments showed the same trend, that is, the C:N and C:P were the highest in DG, and it was the lowest in LT. The reason for this may be the N concentration of LT was the highest, thus reducing its ratio. Relevant studies showed that the C:N of grass in China is usually around 16 [42]. The C:N of the four treatments in this study was above 19, which was higher than the average level of Chinese herbs, and this is consistent with the existing research results [43]. Some scholars believed that the lower the C:P in plants, the higher the growth rate, which has 1:250 as the dividing line, but is generally not lower than 1:250 [44,45], which is consistent with the earlier results of Song et al. [46]. Plant N:P can be used to reveal N limitation or P limitation in the ecosystem [47,48]. It was suggested that plant growth was limited by N when plant N:P < 10 or 14, and plant growth was limited by P when plant N:P > 16 or 20, with different threshold values reported in different grasslands [49,50]. The N:P of plants in the four treatments in this study was greater than 37, exceeding the average value of N:P in the herbaceous plants of China (13.5) [51]. This indicated that the plant growth of grassland ecosystems was limited by P and that the LT was the most severely limited, while NG was relatively lightly limited, and this is consistent with the findings of Song et al. [46]. The reason for this phenomenon may be relevant to the high N concentration and P mineralization rate of soil in LT, which relatively reduced the N:P. Based on the above results, we suggested that appropriate phosphate fertilizer should be added in grassland management in the future to improve the current situation of P limitation.

*4.2. Effects of Different Restoration Modes on Soil C, N and P Concentrations and Ecological Stoichiometry in Grassland Ecosystems*

Plants provide rich substrates for soil through litter and root deposition, so the changes in plant C, N and P concentration will affect soil C, N and P concentration and ecological stoichiometry [52]. In this study, the soil C, N and P of four different restoration measures was different. The SOC concentration of grassland under three artificial restorations was significantly higher than that of natural restoration, among which DG, LP and LT were 1.41, 1.48 and 2.02 times the NG, respectively, indicating that artificial restoration measures are more effective in the SOC accumulation. This may be related to more abundant plant litter and root exudates under artificial restoration measures, which is consistent with Zhang et al. [53]. However, some studies suggested that the soil SOC concentration in the natural grassland was higher than the artificial grassland [54], which may be connected to the study area and soil background. As an essential mineral nutrient for plant growth and development, the soil N mainly comes from litter, rhizobia and biological nitrogen fixation, as well as N deposition in the atmosphere [55]. In this study, the TN concentration reached 2.14 g/kg in LT, which was 2.71, 1.40 and 1.29 times the NG, DG and LP, respectively. The soil TN concentration was the highest in LT and the lowest in NG. This may be relevant to the nitrogen fixation of white clover rhizobia in LT, which is consistent with Xing et al. [54]. Litter and rock weathering are the main sources of soil P [56]. In this study, the TP concentration in three artificial restoration measures was 1.17, 1.3 and 1.70 times that in the natural restoration measures, respectively. This indicated that artificial restoration can improve the phosphorus concentration of the soil. What is more, the promotion effect in LT was the most obvious, which is consistent with the research results of degraded grasslands in the Qinghai–Tibet Plateau [57]. This may be connected to the acceleration of soil P mineralization by high productivity.

Soil C:N, C:P and N:P are important indicators to measure soil quality, and the balance of their ecological stoichiometry is of great significance for maintaining ecosystem function and stability [58]. Soil C:N is an vital index to measure soil nitrogen mineralization capacity. Generally, the larger the C:N, the more unfavorable it is for N mineralization, nutrient release, plant absorption and utilization [59]. The C:N of NG, DG, LP and LT were 15.91, 11.68, 11.71 and 12.02, respectively, which were close to the mean values of 12.3 in the soil of China [60] and 11.83 in the global grasslands [61]. The C:N was significantly highest in NG, indicating that the N mineralization and nutrient release rate of the three artificial restoration measures were stronger than that of the natural restoration measures, which is consistent with the research results from the Qinghai–Tibet Plateau [54]. Soil C:P is an crucial index to measure the phosphorus mineralization capacity for soil, which can indicate the potential of releasing phosphorus or absorbing and retaining phosphorus in the process of organic matter mineralization [62]. The lower the C:P, the higher the availability of soil phosphorus, and the higher the C:P, the less nutrients are released during the decomposition of organic matter [63]. The C:P in NG, DG, LP and LT were 15.36, 18.69, 16.92 and 18.55, respectively, which were far lower than the mean values in the soil of China (52.64) [60] and the global grasslands (64.26) [61]. This indicated that the soil phosphorus availability of natural restoration measures in this study were relatively higher, and this is consistent with the research results from the Qinghai–Tibet Plateau [54]. Soil N:P is an important index to judge the limitation of N and P in soil, which can reflect the threshold value of nutrient element limitation [63]. The soil N:P in DG, LP and LT was significantly higher than NG, but still far lower than the mean values in the soil of China (4.2) [60] and global grasslands (5.55) [61]. According to previous research results, plant growth was limited by P when N:P was greater than 16, and the plant growth was limited by N when N:P was less than 14 [64,65]. Therefore, plant growth may be limited by N in the four restoration measures, which is consistent with the findings of Song et al. [12] and with similar studies in karst areas [66–68]. In order to achieve the nutrient balance and sustainable development of grasslands, appropriate N fertilizer should be added and

N-fixing plants could be seeded to alleviate N limitation in grassland management in the future.

### 4.3. Effects of Different Restoration Modes on MBC, MBN and MBP Concentrations and Ecological Stoichiometry in Grassland Ecosystems

As an important component of soil ecosystems, microorganisms are sensitive to changes in external environment [69], and are an ideal indicator to measure the change in soil quality, playing a direct driving role in maintaining ecosystem function [70] and stability [71]. In this study, the MBC:MBN of the four treatments ranged from 2.85 to 4.82, which was lower than the global average value (8.20) [61]. The MBC, MBN and MBP concentrations were significantly highest in LT, which may be due to the higher concentrations of SOC, TP and TN in LT. So, it provided a better environment for the growth of soil microorganisms, accelerated the circulation of C, N and P in soil and promoted the decomposition of C, N and P by microorganisms [72]. The ecological stoichiometric ratios of MBC:MBN, MBC:MBP and MBN:MBP can reflect soil nutrient level and soil microbial state, and predict changes in the microbial community structure [61]. The MBC:MBN ratio in LT was significantly lower than NG. This may be caused by the fact that the planting of leguminous forage in LT provided rhizobia to promote nitrogen fixation, and then increased the concentration of MBN, thus reducing the ratio of MBC and MBN, which is consistent with previous studies [73]. There were significant differences in MBC:MBP among the four treatments; NG and DG were significantly higher than LP and LT. This may be because, when the phosphorus concentration can meet the requirements of microorganisms, it will improve the utilization rate of microorganisms on carbon, thus leading to a reduction in the assimilation and absorption capacity of microorganisms in terms of total carbon, which is consistent with the findings of Zhang et al. [74]. The MBN:MBP in this study ranged from 5.56 to 6.22, slightly lower than the global mean value (6.9) [75], but higher than the mean value of the desert grasslands of China (0.20) [76]. This may be due to the enhancement of soil microbial nitrogen immobilization and phosphorus mineralization by artificial restoration measures, which increased soil MBN:MBP, and this is consistent with the findings of Dong et al. [77].

### 4.4. Correlation Analysis of C, N and P Concentrations and Ecological Stoichiometry in Plant–Soil–Microbial of Grassland Ecosystems

By simplifying the complex ecological process to the quantitative relationship and dynamic balance between the basic compositional elements of matter, ecological stoichiometry theory can unify the research results of different levels (e.g., plant–soil–microorganism) from the perspective of element ratios for systematic analysis [78]. The results showed that the SOC was highly significantly correlated with plant C and MBC, the TN was highly significantly correlated with plant N and MBN, and the TP was highly significantly correlated with plant P and MBP. This indicated that the supply of C, N and P in the soil had a great influence on the nutrient uptake of plants and soil microorganisms, and the cycling processes of C, N and P in the plant–soil–microorganism system were coupled and influenced each other. The C, N and P concentrations of the plant–soil–microorganism system in grasslands under different restoration measures are different to some extents. This may be due to the change in plant–soil–microorganism habitat caused by differences in forage species and succession changes, which eventually act on the C, N and P concentrations and ecological stoichiometry of the plant–soil–microorganism system, which is consistent with the research results of Lu et al. in the Ningxia typical grassland area [79]. The main source of soil C, N, and P is the mineralization of soil organic matter, which, together with the feedback regulation by plants and microorganisms, resulted in a significant correlation between soil C:N, C:P and N:P for different restoration measures. The same significant correlations were found for C, N, and P levels of plants and microorganisms, and the results are consistent with the research carried out by Song et al. in the desert steppe [80].

## 5. Conclusions

In this paper, the natural restoration grassland and artificial grassland in the karst desertification control area of Southwest China were taken as the research object, and we analyzed the effects of different restoration modes on the C, N and P concentrations and ecological stoichiometry of the plant–soil–microorganism system in grasslands, and the relationship between them. We concluded that: (1) compared with natural restoration (NG), the mixed–seed grassland (LT) could significantly increase the C, N and P concentrations of plants, soils and microorganisms, but the plant C:P ratio was the lowest in LT. (2) The C:N ratio of plant and soil indicated that the N mineralization and nutrient release rate of the three artificial restoration measures (DG, LP and LT) were stronger than that of the natural restoration measure (NG). The plant growth of NG was most seriously limited by nitrogen, while the N:P ratio of plant and soil reflected that the plant growth of the four treatments was limited by P. The results showed that the effect of artificial restoration was better than that of natural restoration for increasing plant, soil and microbial nutrient concentrations, especially in the case of the mixed-seed of leguminosae and gramineae forage. However, in future grassland management, appropriate N fertilizer or N-fixing plants and P fertilizer should be considered to improve the situation of N and P limitation for plant growth. Because this study covered only a short period of time, further studies should strengthen the long-term scale in the future to reveal the mechanism of artificial and natural grassland restoration measures on C, N and P, and its stoichiometric relationships in the plant–soil–microorganism system, in order to provide theoretical guidance for the vegetation restoration of the degraded ecosystems in the karst desertification area.

**Author Contributions:** Conceptualization, K.X. and S.S.; methodology, S.S. and Y.C.; software, S.S. and Y.C.; validation, Y.C., S.S. and K.X.; formal analysis, S.S. and Y.C.; investigation, K.X.; resources, Y.C.; data curation, Y.C.; writing—original draft preparation, K.X.; writing—review and editing, S.S.; visualization, S.S.; supervision, Y.C. and K.X.; project administration, Y.C.; funding acquisition, Y.C. and K.X. All authors have read and agreed to the published version of the manuscript.

**Funding:** This research was funded by the Key Science and Technology Program of Guizhou Provence: the Poverty Alleviation Model and Technology demonstration for Ecoindustries Derived from the karst desertification control (No. 5411 2017 QKHPTRC); the Natural Science Research Project of Education Department of Guizhou Province [Qianjiaohe KY Zi (2022) 157]; the Academic New Seedling Fund Project of Guizhou Normal University (Qianshi Xinmiao B15).

**Data Availability Statement:** Not applicable.

**Acknowledgments:** All the authors thank Jinzhong Fang, Cheng He, Shuyu He and Wenfang Zhang for their contributions to the data collection. All the authors also thank the reviewers and editor for their insightful comments and constructive suggestions.

**Conflicts of Interest:** The authors declare no conflict of interest.

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
