# Peer review of "Ecological Stoichiometric Characteristics of Plant–Soil–Microorganism of Grassland Ecosystems under Different Restoration Modes in the Karst Desertification Area"

_agronomy, doi:10.3390/agronomy13082016_

Round 1

Reviewer 1 Report

Comments

The manuscript primarily focuses on examining the ecological stoichiometric characteristics of the plant-soil-microorganism ecosystem in grasslands under different restoration modes in a karst desertification area. While the research topic is of some importance, the methodology used appears to be relatively straightforward. Furthermore, the limited amount of information gathered at the sampling points weakens the overall conclusion of the article. Therefore, it is recommended that the author make some modifications to the methodology and data collection process to ensure the validity and reliability of the results.

1.      The spatial map of the research area, as well as the exact location of the sampling points, including the latitude and longitude, need to be presented in detail in the research methods section. Additionally, the location of the sampled photos and the specific amount of data collected must also be included. However, the author has not yet provided these important details, which are crucial for understanding and replicating the study. Therefore, it is necessary for the author to thoroughly modify the research methods section to include all of these pertinent details in order for the study to be considered authentic and reliable. This is crucial for establishing the credibility and reliability of the research.

2.      The manuscript only utilized a single period of data to draw conclusions, which is insufficient. In order to establish the validity of the findings, multiple years of data and multi-period sampling data are necessary.

3.      Furthermore, it is unclear if the grassland restoration period is consistent across all sampling points. Has the author conducted further investigations to confirm this? This information is critical to understanding the overall effectiveness of grassland restoration efforts and should be thoroughly addressed in the research methods section.

4.      Regarding the terminology used in the article, it is recommended that the author use "Restoration modes" instead of "Restoration models." Additionally, it is highly recommended that the author thoroughly review the entire text and carefully revise the English version to ensure accuracy and clarity.

5.      There is a low-level error in the article where the titles of sections 4.1 and 4.2 are duplicated. This error is unacceptable and reflects poorly on the overall quality of the research. Therefore, it is strongly advised that the author carefully review the entire text to identify and correct any such errors.

The language used throughout the manuscript should also be reviewed carefully to ensure that it is written in native and authentic research English. 

Author Response

Dear Editor and Reviewers, Thank you for your letter and the reviewers’ comments concerning our manuscript entitled “Ecological Stoichiometric Characteristics of Grassland Ecosystem Plant-Soil-Microorganism under Different Restoration Models in the Karst Desertification Area” (ID: 2495615). Those comments are valuable and very helpful. We have read through comments carefully and have made corrections. Based on the instructions provided in your letter, we uploaded the file of the revised manuscript. Revisions in the text are shown using red highlight. The responses to the reviewer’s comments are marked in red and presented following. We would love to thank you for allowing us to resubmit a revised copy of the manuscript and we highly appreciate your time and consideration. Best wishes, Kangning Xiong, Shuzhen Song. Reviewer 1 The manuscript primarily focuses on examining the ecological stoichiometric characteristics of the plant-soil-microorganism ecosystem in grasslands under different restoration modes in a karst desertification area. While the research topic is of some importance, the methodology used appears to be relatively straightforward. Furthermore, the limited amount of information gathered at the sampling points weakens the overall conclusion of the article. Therefore, it is recommended that the author make some modifications to the methodology and data collection process to ensure the validity and reliability of the results. 1.    The spatial map of the research area, as well as the exact location of the sampling points, including the latitude and longitude, need to be presented in detail in the research methods section. Additionally, the location of the sampled photos and the specific amount of data collected must also be included. However, the author has not yet provided these important details, which are crucial for understanding and replicating the study. Therefore, it is necessary for the author to thoroughly modify the research methods section to include all of these pertinent details in order for the study to be considered authentic and reliable. This is crucial for establishing the credibility and reliability of the research. Response: We are extremely grateful to reviewer for pointing out this problem, we have added the spatial map of the research area, the exact location of the sampling points and the location of the sampled photos in the study area and sample plot setting section, please see Figure 1 in the revised manuscript. While the specific amount of data collected were presented in the Sample collection section. We have supplemented the methodology section thoroughly, including as much detail as possible. Thanks again. 2.    The manuscript only utilized a single period of data to draw conclusions, which is insufficient. In order to establish the validity of the findings, multiple years of data and multi-period sampling data are necessary. Response: We agree with the reviewer’s that it is necessary to also present multiple years of data and multi-period sampling data to establish the validity of the findings. Thank you for your precious advice. This comment is valuable and very helpful for revising and improving our paper. We also recognized this deficiency. However, in this work, we tried to investigate the effects of different vegetation restoration modes on the ecological stoichiometry of plant-soil-microorganism ecosystem in grasslands of karst desertification area to reveal the current situation of grassland development. We do like that our findings presented here are of significance to grassland management, but multiple years of data and multi-period sampling data requires some further development. We will therefore present multiple years of data and multi-period sampling data in our future work, moreover, we have added it as research deficiency in the conclusion section of this paper in the revised manuscript.Thanks again. 3.    Furthermore, it is unclear if the grassland restoration period is consistent across all sampling points. Has the author conducted further investigations to confirm this? This information is critical to understanding the overall effectiveness of grassland restoration efforts and should be thoroughly addressed in the research methods section. Response: Thank you for your precious comments and advice. We have added this information in the research methods section in the revised manuscript as following: In this study, all the land was planted with corn before planting grass and natural restoration. After the implementation of rocky desertification control project, the land use mode changed from corn planting to artificial grass establishment or returning farmland to grassland (natural restoration). 4.    Regarding the terminology used in the article, it is recommended that the author use "Restoration modes" instead of "Restoration models." Additionally, it is highly recommended that the author thoroughly review the entire text and carefully revise the English version to ensure accuracy and clarity. Response: We deeply appreciate the reviewer’s suggestion, we have used “Restoration modes” instead of “Restoration models” in the entire text in the revised manuscript. At the same time, we have thoroughly reviewed the entire text and carefully revised the English version to ensure accuracy and clarity. 5.     There is a low-level error in the article where the titles of sections 4.1 and 4.2 are duplicated. This error is unacceptable and reflects poorly on the overall quality of the research. Therefore, it is strongly advised that the author carefully review the entire text to identify and correct any such errors. Response: We apologize for the confusion caused by the close proximity of the titles of sections 4.1 and 4.2. The titles of sections 4.1 is “Effects of different restoration modes on plant C, N, P concentrations and ecological stoichiometry in grassland ecosystem”, and the titles of sections 4.2 is “Effects of different restoration modes on soil C, N, P concentrations and ecological stoichiometry in grassland ecosystem”. Sorry again. Comments on the Quality of English Language The language used throughout the manuscript should also be reviewed carefully to ensure that it is written in native and authentic research English.  Response: We apologize for the language problems in the original manuscript. The language presentation was improved with assistance from a native English speaker with appropriate research background.

Reviewer 2 Report

 This is a straightforward analysis of C, N and P in three planted pasture treatments and a naturally restored grassland. Nutrients were analyzed in plants, soils, and microorganisms, and stoichiometric ratios were calculated. The statistical analyses are appropriate. The Discussion compares these ratios with published values, explains their ecological significance, and makes recommendations for grassland management. The management recommendations are to fertilize with N and P, but for N could include planting legumes, as the legume treatment showed improved N stoichiometry.

Additional information could be provided about research site and the restoration treatments. Were these lands grazed in the past, or subject to tree harvesting? Are Lolium perenne, Dactylis glomerata, and Trifolium repens indigenous to these grasslands, or do they originate in other regions of China, Asia, or Europe? Explain natural restoration—I assume this means allowing native vegetation to reestablish after grazing animals are excluded, but please be specific.

Monoculture planted grassland may be more productive for grazing, but native grassland has higher biodiversity. Since biodiversity loss is not considered here, the authors should write that the goal of this study is to improve grasslands for grazing.

I have a number of specific suggestions for revision to improve all sections of the manuscript, as well as language and sentence structure:

17 spell out MBC, MBN and MBP first time.

Omit (P < 0.05) in the Abstract. You do not need to specify significance level in the Abstract.

35 “ play a key role in nutrient cycle, ecosystem process and function”

CHANGE TO

 play a key role in nutrient cycles, ecosystem processes and functioning 

36 Plant-soil-microorganism is a closely related organism.”

This is incorrectly stated, as “Plant-soil-microorganism” is NOT an organism. Perhaps you mean that nutrients within the plant-soil-microorganism system are regulated by flows within the components?

41  “construction” I think you mean composition.

 44 I don’t believe “evolution” is the correct word here. Perhaps you mean productivity and stability?

44 “So it is helpful to reveal the nutrient cycle law and system stability mechanism of ecosystems by studying the internal relation-ship of ecosystem processes and their feedback effects with plant-soil-microorganism C, N and P as carriers [6,7].”

The term “law” is rarely applied to nutrient cycling. If you wish to use “law”, you need to provide a citation.

I suggest the following revision:

So it is helpful to reveal nutrient cycling processes and mechanisms of stability by studying the internal relationships of ecosystem processes and their feedback effects within the plant-soil-microorganism system, with a focus on C, N and P as carriers [6,7].

49 “material cycle of each component of ecosystem”

CHANGE TO

material cycles of each component of the ecosystem 

50 “research of biogeochemical cycle”

CHANGE TO

research of biogeochemical cycles 

51 “As an important component unit of terrestrial 51 ecosystem,”

CHANGE TO

As an important component of terrestrial 51 ecosystems, 

52 “of ecosystem”

CHANGE TO

of ecosystems OR of the ecosystem

53 “of plant community and the dynamic change of soil micro- organism”

CHANGE TO

of plant communities and dynamic changes of soil micro-organisms 

57 “Currently, it has a lot of studies on ecological stoichiometry characteristics of degraded ecosystems  involving a wide range of research areas and objects “

CHANGE TO

Currently, there are numerous studies on ecological stoichiometry of degraded ecosystems involving a wide range of research areas and objectives

60 “of plant-soil- microorganism under different regions …”

CHANGE TO

of plant-soil-microorganism systems in different regions…

70 “which is the largest and most concentrated and contiguous karst ecological fragile area among the three karst concen-trated distribution areas in the world, and its exposed area is more than …”

CHANGE TO

which is the largest and most contiguous fragile karst area among the three extensive karst distribution areas in the world, with a total area of more than …

74, 77 “task” is not the correct word here. Do you mean rocky desertification is the main perturbation?

You need to explain what rocky desertification is. What was the soil surface like before desertification? Was it sand or loam with plant cover? Did grazing and erosion lead to rocky desertification?

81 Explain what you mean by “natural restoration of grassland”  Do you mean natural recovery and secondary succession without human intervention?

85 “many achievements in forest ecosystem, vegetation succession or regional type” Unclear. What do you mean by regional type? Is the forest ecosystem a regional type? Or should this read “other regional vegetation types”?

89 What are “variation law”  and “internal correlation” Explain and use a citation.

91 “was taken the grassland ecosystem”

CHANGE TO

was done in the grassland ecosystem 

95, 97, and other lines throughout manuscript “plant-soil-microorganism”

CHANGE TO plant-soil microorganism system, OR plants, soils, and microorganisms

108 “accounting for 64.93% of the demonstration area”

What is the demonstration area? Please define.

Change to 64.9%. I imagine the scale of measurement is not precise to the nearest 0.01 percent.

111 “The soil type is mainly yellow soil, and there are also other types such as calcareous soil.”

Since this is a karst landscape, isn’t all of the soil calcareous? Karst is defined as a landscape underlain by limestone (which is calcareous). Is the yellow soil also calcareous? Please clarify.

112 “arbor-shrub vegetation”

CHANGE TO

tree-shrub vegetation 

119 What is the “karst desertification control project”. How was the land disturbed initially? Tree cutting? Grazing? Agriculture? If available, provide a reference to this project.

120 “natural restoration grassland”

again, explain what you mean by natural grassland restoration. Was vegetation allowed to recover naturally after cessation of grazing, tree cutting, agriculture…? Did trees and shrubs recolonize as well as grasses and forbs? How many species of grasses and forbs occurred in the natural grassland restoration treatment?

133 “shall be set in each plot……shall be greater….”

CHANGE TO

were set in each of the four restoration treatments….was greater…

134 “All the plant in each plot were cut to form mixed into one replicate sample”

CHANGE TO

All the plants in each 1mx1m quadrat were clipped and mixed into one replicate sample…

138 “Remove the impurities in the soil sample”

Do you mean rocks and large organic debris were removed?

144 “After drying, the samples were sieved with a 2mm sieve for the determination of plant carbon, nitrogen and phosphorus”  Plant samples are normally ground to a fine powder using a ball grinder or a Wiley mill, to homogenize them prior to nutrient analysis. You need to add this to your description of the methods for plant analysis.

161 The term “content” or “contents” should be changed to concentration or concentrations throughout the manuscript. Concentration = mass of nutrient/unit plant [or soil or microbial] mass. Content = (mass of nutrient/unit plant mass) X total plant mass

Figure 1 Define the abbreviations NG, DG, LP, and LT in the legend so the figure stands alone. Also define abbreviations in other figure legends.

Do you also have total plant biomass (g/m2) data to report? If this was published elsewhere, please reference the study and explain the biomass patterns; if not, please report biomass data here. The biomass data may be useful to interpret your results.

220 “SOC was severely significant positive correlated with TN,….”

CHANGE TO

SOC was highly significantly positively correlated with TN, 

Make this change throughout the manuscript, and replace “severely significant” with highly significant

Because this sentence structure has three adjectives in sequence and is quite clumsy, you might try rewording some of the sentences. For instance,

The positive correlation of SOC was highly significant with TN, …… There was a highly significant negative correlation with plant C:N, and a significantly negative correlation with plant C:P, MBC:MBN.   Etc.

Figure 4 is difficult to read because the type is small and faint. It is up to the Editor, whether this needs revision.

259 “C, N and P is the basic elements”

CHANGE TO

C, N and P are basic elements   [remove “the” because there are additional basic elements]

274 “The main reason for t may”

CHANGE TO

The main reason for this may

276 “…desert steppe area [37], which may be related…”

Long sentence lines 265-270. Break into two sentences:

…desert steppe area [37]. This may be related….

274 “main reason for t may be…”

CHANGE TO

main reason for this may be

275 What is “G”?

276 “which was indicated”

CHANGE TO

which indicated

286-290 This sentence is very long and redundant. Break into two sentences and shorten.

294 “….N:P<10 or 14,, and plant growth was limited by P when plant N:P>16 or 20 [49,50].“

The range of N:P values (10 or 14; 16 or 20) is unclear. Were these from different publications [49, 50]? If that is the case, you could clarify by explaining: ‘with different threshold values reported in different grasslands [49,50}.’

290 “Some scholars believe that the lower the C:P in plants was lower, and the growth rate and the primary productivity was the higher [44,45],”

This sentence is unclear. Include the C:P values in the sentence to illustrate the lower and higher values you are referring to?

322-327 Very long sentence—break into two sentences.

330 “for resisting global change” [58].

Be specific:  do you mean for sequestering atmospheric C to ameliorate global warming?

355 “appropriate N fertilizer should be increased”

Could you add: N-fixing plants could be seeded?

359 “microorganism is sensitive to changes 359 in external environment [69], it is an ideal indicator… ”

CHANGE To

microorganisms are sensitive to changes in the external environment [69], and are an ideal indicator….

363-367 sentence is too long. Break into two sentences.

CHANGE TO

and we analyzed the effects of different restoration models 

410 “plant-soil-microorganism in grassland ecosystem,”

CHANGE TO

plant-soil-microorganism system in grasslands,

Make this change throughout the manuscript

418 “the current four treatments was limited by P element”

CHANGE TO

all four treatments were limited by P.

419 “artificial restoration was better than that of natural restoration”

add: for increasing plant, soil and microbial nutrient concentrations.

I made many edits to improve the English language and sentence structure. I have not corrected all language issues. I suggest the authors ask a colleague whose first language is English to help them edit the manuscript to improve the English.

Author Response

Dear Editor and Reviewers,

Thank you for your letter and the reviewers’ comments concerning our manuscript entitled “Ecological Stoichiometric Characteristics of Grassland Ecosystem Plant-Soil-Microorganism under Different Restoration Models in the Karst Desertification Area” (ID: 2495615). Those comments are valuable and very helpful. We have read through comments carefully and have made corrections. Based on the instructions provided in your letter, we uploaded the file of the revised manuscript. Revisions in the text are shown using red highlight. The responses to the reviewer’s comments are marked in red and presented following.

We would love to thank you for allowing us to resubmit a revised copy of the manuscript and we highly appreciate your time and consideration.

Best wishes,

Kangning Xiong, Shuzhen Song.

Reviewer 2

 This is a straightforward analysis of C, N and P in three planted pasture treatments and a naturally restored grassland. Nutrients were analyzed in plants, soils, and microorganisms, and stoichiometric ratios were calculated. The statistical analyses are appropriate. The Discussion compares these ratios with published values, explains their ecological significance, and makes recommendations for grassland management. The management recommendations are to fertilize with N and P, but for N could include planting legumes, as the legume treatment showed improved N stoichiometry.

Response:We agree with the comment. Management recommendations for planting legumes to improve N limitation were added to the appropriate locations in the revised manuscript.

Additional information could be provided about research site and the restoration treatments. Were these lands grazed in the past, or subject to tree harvesting?

Response: Thank you for your precious comments and advice. We have added this information in the research methods section in the revised manuscript as following: In this study, all the land was planted with corn before planting grass and natural restoration. After the implementation of rocky desertification control project, the land use mode changed from corn planting to artificial grass establishment or returning farmland to grassland (natural restoration).

Are Lolium perenne, Dactylis glomerata, and Trifolium repens indigenous to these grasslands, or do they originate in other regions of China, Asia, or Europe? Explain natural restoration—I assume this means allowing native vegetation to reestablish after grazing animals are excluded, but please be specific.

Response: We are grateful for the suggestion and added the variety information of Lolium perenne, Dactylis glomerata, and Trifolium repens in the revised manuscript as the following: The variety of Trifolium repens is “Haifa”, the variety of Dactylis glomerata is “Qiangrass No. 4”, and the variety of Lolium perenne is “Yaqing”, and the three kinds of forage seeds are provided by Guizhou Shennong Seed Industry Co., Ltd. and Lvyi Seed Industry Co., Ltd. In addition, we have added the explanation of natural restoration in the revised manuscript as the following: After the implementation of rocky desertification control project, the land use mode changed from corn planting to artificial grass establishment or returning farmland to grassland (natural restoration).

Monoculture planted grassland may be more productive for grazing, but native grassland has higher biodiversity. Since biodiversity loss is not considered here, the authors should write that the goal of this study is to improve grasslands for grazing.

Response: Thank you for your comments, we have added the goal of this study in the revised manuscript as the following: According to the layout of rocky desertification control project, and considering the development of herbivorous animal husbandry, the research group established artificial grassland and returned farmland to grassland in the study area in 2012.

I have a number of specific suggestions for revision to improve all sections of the manuscript, as well as language and sentence structure:

 17 spell out MBC, MBN and MBP first time.

Response:We are grateful for the suggestion and spelled out MBC, MBN and MBP in the Abstract section of revised manuscript.

Omit (P < 0.05) in the Abstract. You do not need to specify significance level in the Abstract.

Response: Thank you for the suggestion. We have removed P < 0.05 in the Abstract section of revised manuscript.

35 “ play a key role in nutrient cycle, ecosystem process and function”

CHANGE TO

 …play a key role in nutrient cycles, ecosystem processes and functioning 

Response: Thank you for the comments, and we have rewrote it in the revised manuscript.

36 “Plant-soil-microorganism is a closely related organism.”

This is incorrectly stated, as “Plant-soil-microorganism” is NOT an organism. Perhaps you mean that nutrients within the plant-soil-microorganism system are regulated by flows within the components?

Response: Thank you for the comments, and we have rewrote it in the revised manuscript as following: Plant-soil-microorganism are closely related and interact with each other.

41  “construction” I think you mean composition.

Response: We are agree with your comments, and we have rewrote it in the revised manuscript.

 44 I don’t believe “evolution” is the correct word here. Perhaps you mean productivity and stability?

Response: Thank you for the comments, and we have rewrote it in the revised manuscript as following: The cycle and accumulation process of nutrients among plants, soil and microorganisms promotes the restoration, productivity and stability of ecosystems.

44 “So it is helpful to reveal the nutrient cycle law and system stability mechanism of ecosystems by studying the internal relation-ship of ecosystem processes and their feedback effects with plant-soil-microorganism C, N and P as carriers [6,7].”

The term “law” is rarely applied to nutrient cycling. If you wish to use “law”, you need to provide a citation.

I suggest the following revision:

So it is helpful to reveal nutrient cycling processes and mechanisms of stability by studying the internal relationships of ecosystem processes and their feedback effects within the plant-soil-microorganism system, with a focus on C, N and P as carriers [6,7].

Response: Thank you for the comments, and we have rewrote it in the revised manuscript as following: So it is helpful to reveal nutrient cycling processes and mechanisms of stability by studying the internal relationships of ecosystem processes and their feedback effects within the plant-soil-microorganism system, with a focus on C, N and P as carriers [6,7].

49 “material cycle of each component of ecosystem”

CHANGE TO

material cycles of each component of the ecosystem 

Response: Thank you for the comments, and we have rewrote it in the revised manuscript.

50 “research of biogeochemical cycle”

CHANGE TO

research of biogeochemical cycles 

Response: Thank you for the comments, and we have rewrote it in the revised manuscript.

51 “As an important component unit of terrestrial 51 ecosystem,”

CHANGE TO

As an important component of terrestrial 51 ecosystems, 

Response: Thank you for the suggestions, and we have rewrote it in the revised manuscript.

52 “of ecosystem”

CHANGE TO

of ecosystems OR of the ecosystem

Response: Thank you for the comments, and we have rewrote it in the revised manuscript.

53 “of plant community and the dynamic change of soil micro- organism”

CHANGE TO

of plant communities and dynamic changes of soil micro-organisms 

Response: Thank you for the suggestions, and we have rewrote it in the revised manuscript. 

57 “Currently, it has a lot of studies on ecological stoichiometry characteristics of degraded ecosystems  involving a wide range of research areas and objects “

CHANGE TO

Currently, there are numerous studies on ecological stoichiometry of degraded ecosystems involving a wide range of research areas and objectives

Response: Thank you for the comments, and we have rewrote it in the revised manuscript. 

60 “of plant-soil- microorganism under different regions …”

CHANGE TO

of plant-soil-microorganism systems in different regions…

Response: Thank you for the suggestions, and we have rewrote it in the revised manuscript. 

 70 “which is the largest and most concentrated and contiguous karst ecological fragile area among the three karst concen-trated distribution areas in the world, and its exposed area is more than …”

CHANGE TO

which is the largest and most contiguous fragile karst area among the three extensive karst distribution areas in the world, with a total area of more than …

Response: Thank you for the comments.,and we have rewrote it in the revised manuscript. 

74, 77 “task” is not the correct word here. Do you mean rocky desertification is the main perturbation?

Response: Thank you for the suggestions. We have rewrote it in the revised manuscript as following: Influenced by geological background and long-term human factors, rocky desertification is the most serious eco-environmental problem in this region [21].

You need to explain what rocky desertification is. What was the soil surface like before desertification? Was it sand or loam with plant cover? Did grazing and erosion lead to rocky desertification?

Response: We are extremely grateful to reviewer for pointing out this problem.We have added the definition of rocky desertification in the revised manuscript as following: Rocky desertification is a landscape phenomenon in which karst landscapes in tropical and subtropical climates are disturbed by the natural environment and humans, resulting in severe soil erosion, vegetation decline, and a desert-like surface [21].

81 Explain what you mean by “natural restoration of grassland”  Do you mean natural recovery and secondary succession without human intervention?

Response: Thank you for the suggestion. We have rewrote it in the revised manuscript as following: The establishment of artificial grassland or natural recovery and secondary succession without human intervention of grassland may change the nutrient relationship of soil-plant-microorganism formed by long-term evolution [23,24].

85 “many achievements in forest ecosystem, vegetation succession or regional type” Unclear. What do you mean by regional type? Is the forest ecosystem a regional type? Or should this read “other regional vegetation types”?

Response: We are extremely grateful to reviewer for pointing out this problem. We have rewrote it in the revised manuscript as following: Up to now, the research on ecological stoichiometry in the karst area has many achievements on vegetation succession, some vegetation (e.g. forest), or areas (e.g. grassland in northern China) [27-31].

89 What are “variation law”  and “internal correlation” Explain and use a citation.

Response: Thank you for the suggestion. This sentence was not accurate enough in the original manuscript, we have made changes in the revised manuscript as following: At the same time, the research on the C, N, P concentration of plant-soil-microorganism and ecological stoichiometry characteristics of grassland ecosystem under different restoration measures in the karst desertification control area is still insufficient.

91 “was taken the grassland ecosystem”

CHANGE TO

was done in the grassland ecosystem 

Response: Thank you for the suggestions, and we have rewrote it in the revised manuscript.

95, 97, and other lines throughout manuscript “plant-soil-microorganism”

CHANGE TO plant-soil microorganism system, OR plants, soils, and microorganisms

Response: Thank you for the comments, and we have rewrote it in the revised manuscript.

108 “accounting for 64.93% of the demonstration area”

What is the demonstration area? Please define.

Change to 64.9%. I imagine the scale of measurement is not precise to the nearest 0.01 percent.

Response: We are extremely grateful to reviewer for pointing out this problem. We wanted to express the study area rather than the demonstration area, and we have revised this in the revised manuscript.

111 “The soil type is mainly yellow soil, and there are also other types such as calcareous soil.”

Since this is a karst landscape, isn’t all of the soil calcareous? Karst is defined as a landscape underlain by limestone (which is calcareous). Is the yellow soil also calcareous? Please clarify.

Response: Thank you for your valuable comments, we have defined the soil type in the study area as yellow soil according to the international soil classification standards, so deleted the sentence of “ and there are also other types such as calcareous soil” in the revised manuscript.

112 “arbor-shrub vegetation”

CHANGE TO

tree-shrub vegetation 

Response: Thank you for the suggestions, and we have rewrote it in the revised manuscript. 

119 What is the “karst desertification control project”. How was the land disturbed initially? Tree cutting? Grazing? Agriculture? If available, provide a reference to this project.

Response: Thank you for the comments. We have added a reference it in the revised manuscript as following : According to the layout of rocky desertification control project, and considering the   development of herbivorous animal husbandry, the research group established artificial grassland and returned farmland to grassland in the study area in 2012 [21].

120 “natural restoration grassland”

again, explain what you mean by natural grassland restoration. Was vegetation allowed to recover naturally after cessation of grazing, tree cutting, agriculture…? Did trees and shrubs recolonize as well as grasses and forbs? How many species of grasses and forbs occurred in the natural grassland restoration treatment?

Response: We are grateful for the suggestion, we have added the explanation of natural restoration in the revised manuscript as the following: After the implementation of rocky desertification control project, the land use mode changed from corn planting to artificial grass establishment or returning farmland to grassland (natural restoration).

133 “shall be set in each plot……shall be greater….”

CHANGE TO

were set in each of the four restoration treatments….was greater…

Response: Thank you for the suggestions, and we have rewrote it in the revised manuscript. 

134 “All the plant in each plot were cut to form mixed into one replicate sample”

CHANGE TO

All the plants in each 1mx1m quadrat were clipped and mixed into one replicate sample…

Response: Thank you for the comments, and we have rewrote it in the revised manuscript. 

138 “Remove the impurities in the soil sample”

Do you mean rocks and large organic debris were removed?

Response: Thank you for the suggestions, and we have rewrote it in the revised manuscript as following: Remove the rocks and large organic debris in the soil sample, and divide the sample into two parts.

144 “After drying, the samples were sieved with a 2mm sieve for the determination of plant carbon, nitrogen and phosphorus”  Plant samples are normally ground to a fine powder using a ball grinder or a Wiley mill, to homogenize them prior to nutrient analysis. You need to add this to your description of the methods for plant analysis.

Response: Thank you for your precious advice.We have added it in the revised manuscript as following: After drying, plant samples were normally ground to a fine powder using a ball grinder and sieved with a 2mm sieve for the determination of plant carbon, nitrogen and phosphorus.

161 The term “content” or “contents” should be changed to concentration or concentrations throughout the manuscript. Concentration = mass of nutrient/unit plant [or soil or microbial] mass. Content = (mass of nutrient/unit plant mass) X total plant mass

Response: Thank you for the suggestions, and we have rewrote it throughout in the revised manuscript. 

Figure 1 Define the abbreviations NG, DG, LP, and LT in the legend so the figure stands alone. Also define abbreviations in other figure legends.

Response:  Thank you for your valuable comments, and we have defined NG, DG, LP, and LT in the legend of all the figures in the revised manuscript.

Do you also have total plant biomass (g/m2) data to report? If this was published elsewhere, please reference the study and explain the biomass patterns; if not, please report biomass data here. The biomass data may be useful to interpret your results.

Response: We agree with the comment, and we have added it in the material and methods section in the revised manuscript as following: The annual dry weight of DG, LP, LT and NG was 1049g/m2, 1593g/m2, 1988g/m2 and 1153g/m2, respectively.

220 “SOC was severely significant positive correlated with TN,….”

CHANGE TO

SOC was highly significantly positively correlated with TN, 

Make this change throughout the manuscript, and replace “severely significant” with highly significant

Because this sentence structure has three adjectives in sequence and is quite clumsy, you might try rewording some of the sentences. For instance,

The positive correlation of SOC was highly significant with TN, …… There was a highly significant negative correlation with plant C:N, and a significantly negative correlation with plant C:P, MBC:MBN.   Etc.

Response: Thank you for the suggestions, and we have rewrote it throughout in the revised manuscript. 

Figure 4 is difficult to read because the type is small and faint. It is up to the Editor, whether this needs revision.

Response: Thank you for finding this deficiency, we have modified the Figure 4 in the revised manuscript.

259 “C, N and P is the basic elements”

CHANGE TO

C, N and P are basic elements   [remove “the” because there are additional basic elements]

Response: Thank you for the suggestions, and we have removed it in the revised manuscript.

274 “The main reason for t may”

CHANGE TO

The main reason for this may

Response: Thank you for the comments, and we have rewrote it in the revised manuscript.

276 “…desert steppe area [37], which may be related…”

Long sentence lines 265-270. Break into two sentences:

…desert steppe area [37]. This may be related….

Response: Thank you for the suggestions, and we have rewrote it in the revised manuscript.

274 “main reason for t may be…”

CHANGE TO

main reason for this may be

Response: Thank you for the comments, and we have rewrote it in the revised manuscript.

275 What is “G”?

Response: Thank you for finding this deficiency, we have rewrote it in the revised manuscript.  

276 “which was indicated”

CHANGE TO

which indicated

Response: Thank you for the suggestions, and we have rewrote it in the revised manuscript.

286-290 This sentence is very long and redundant. Break into two sentences and shorten.

294 “….N:P<10 or 14,, and plant growth was limited by P when plant N:P>16 or 20 [49,50].“

The range of N:P values (10 or 14; 16 or 20) is unclear. Were these from different publications [49, 50]? If that is the case, you could clarify by explaining: ‘with different threshold values reported in different grasslands [49,50}.’

 Response: We are grateful for the suggestion, and we have rewrote it in the revised manuscript.

290 “Some scholars believe that the lower the C:P in plants was lower, and the growth rate and the primary productivity was the higher [44,45],”

This sentence is unclear. Include the C:P values in the sentence to illustrate the lower and higher values you are referring to?

Response: We agree with the comment, and we have added the the C:P values in the sentence in the revised manuscript as following: Some scholars believe that the lower the C:P in plants, the higher the growth rate, which is 1:250 as the dividing line, but generally not lower than 1:250 [44,45].

 322-327 Very long sentence—break into two sentences.

 Response: We are grateful for the suggestion, and we have rewrote it in the revised manuscript.

330 “for resisting global change” [58].

Be specific:  do you mean for sequestering atmospheric C to ameliorate global warming?

Response: Thank you for the suggestion, and we have rewrote it in the revised manuscript as following: Soil C:N, C:P and N:P are important indicators to measure soil quality, and the balance of their ecological stoichiometry is of great significance for maintaining ecosystem function and stability [58].

355 “appropriate N fertilizer should be increased”

Could you add: N-fixing plants could be seeded?

 Response: We are grateful for the suggestion, and we have added it in the revised manuscript.

359 “microorganism is sensitive to changes 359 in external environment [69], it is an ideal indicator… ”

CHANGE To

microorganisms are sensitive to changes in the external environment [69], and are an ideal indicator….

 Response: We are grateful for the suggestion, and we have rewrote it in the revised manuscript.

363-367 sentence is too long. Break into two sentences.

CHANGE TO

and we analyzed the effects of different restoration models 

Response: Thank you for your comment, and we have rewrote it in the revised manuscript. 

410 “plant-soil-microorganism in grassland ecosystem,”

CHANGE TO

plant-soil-microorganism system in grasslands,

Make this change throughout the manuscript

Response: we are agree with your comment, and we have rewrote it in the revised manuscript. 

418 “the current four treatments was limited by P element”

CHANGE TO

all four treatments were limited by P.

Response: we are agree with your comment, and we have rewrote it in the revised manuscript. 

419 “artificial restoration was better than that of natural restoration”

add: for increasing plant, soil and microbial nutrient concentrations.

Response: we are agree with your comment, and we have added it in the revised manuscript.  

I made many edits to improve the English language and sentence structure. I have not corrected all language issues. I suggest the authors ask a colleague whose first language is English to help them edit the manuscript to improve the English.

Response: We apologize for the language problems in the original manuscript. The language presentation was improved with assistance from a native English speaker with appropriate research background.
